# Chalcones as Anti-Glioblastoma Stem Cell Agent Alone or as Nanoparticle Formulation Using Carbon Dots as Nanocarrier

**DOI:** 10.3390/pharmaceutics14071465

**Published:** 2022-07-14

**Authors:** Eduardo A. Veliz, Anastasiia Kaplina, Sajini D. Hettiarachchi, Athina L. Yoham, Carolina Matta, Sabrin Safar, Meghana Sankaran, Esther L. Abadi, Emel Kirbas Cilingir, Frederic A. Vallejo, Winston M. Walters, Steven Vanni, Roger M. Leblanc, Regina M. Graham

**Affiliations:** 1Department of Chemistry, University of Miami, 1301 Memorial Drive, Coral Gables, FL 33146, USA; eveliz@mdc.edu (E.A.V.); priobengals@gmail.com (A.K.); sdh86@miami.edu (S.D.H.); ela84@miami.edu (E.L.A.); exk444@miami.edu (E.K.C.); rml@miami.edu (R.M.L.); 2Department of Natural Sciences, Miami Dade College, 300 NE Second Ave, Miami, FL 33132, USA; 3Department of Neurosurgery, University of Miami Miller School of Medicine, 1095 NW 14th Terrace, Miami, FL 33136, USA; a.yoham@umiani.edu (A.L.Y.); cm5786@nyu.edu (C.M.); ssafar@usc.edu (S.S.); mxs2250@miami.edu (M.S.); fav8@med.miami.edu (F.A.V.); wwalters@med.miami.edu (W.M.W.); svanni@med.miami.edu (S.V.); 4University of Miami Brain Tumor Initiative, Department of Neurosurgery, University of Miami Miller School of Medicine, 1095 NW 14th Terrace, Miami, FL 33136, USA; 5HCA Florida University Hospital, 3476 S University Dr., Davie, FL 33328, USA; 6Department of Medicine, Dr. Kiran C. Patel College of Allopathic Medicine, Davie, FL 33328, USA; 7Sylvester Comprehensive Cancer Center, University of Miami Miller School of Medicine, 1475 NW 12th Ave, Miami, FL 33136, USA

**Keywords:** glioblastoma, nanocarrier, carbon dot, chalcone, drug delivery system, glioblastoma stem cell, GSC

## Abstract

The current prognosis for glioblastoma is dismal. Treatment-resistant glioblastoma stem cells (GSCs) and the failure of most drugs to reach therapeutic levels within the tumor remain formidable obstacles to successful treatment. Chalcones are aromatic ketones demonstrated to reduce malignant properties in cancers including glioblastoma. Nanomedicines can increase drug accumulation and tumor cell death. Carbon-dots are promising nanocarriers that can be easily functionalized with tumor-targeting ligands and anti-cancer drugs. Therefore, we synthesized a series of 4′-amino chalcones with the rationale that the amino group would serve as a “handle” to facilitate covalent attachment to carbon-dots and tested their cytotoxicity toward GSCs. We generated 31 chalcones (22 4′-amino and 9 4′ derivatives) including 5 novel chalcones, and found that 13 had an IC_50_ below 10 µM in all GSC lines. After confirming that the 4-amino group was not part of the active pharmacophore, chalcones were attached to transferrin-conjugated carbon-dots. These conjugates were significantly more cytotoxic than the free chalcones, with the C-dot-transferrin-2,5, dimethoxy chalcone conjugate inducing up to 100-fold more GSC death. Several of the tested chalcones represent promising lead compounds for the development of novel anti-GSC drugs. Furthermore, designing amino chalcones for carbon-dot mediated drug delivery is a rational and effective methodology.

## 1. Introduction

The prognosis for individuals diagnosed with Glioblastoma (GBM), the most common and lethal primary brain tumor in adults, remains dismal. Even with an aggressive standard of care that includes; maximal safe tumor resection, radiation therapy with concomitant chemotherapy (temozolomide), the average survival is only 14–15 months [1]. Furthermore, extensive research aimed at understanding the genetic and epigenetic drivers of the disease and thousands of clinical trials have not substantially improved patient outcomes in decades. Treatment failure has been attributed to the inability to fully resect the tumor due to the highly invasive nature of GBM, the lack of effective chemotherapies able to cross the blood–brain barrier (BBB) at therapeutic levels and the presence of treatment resistant GBM stem-like cells (GSCs), which lead to tumor recurrence and patient relapse.

Although GSCs, referred to as GBM stem-like cells or tumor initiating cells, represent only a small population of cells within the tumor, they have the capacity to divide asymmetrically, giving rise to additional GSCs as well as the more differentiated cells that make up the bulk of the tumor [2]. Emerging research indicates that GSCs drive tumor growth by promoting blood vessel formation through vascular endothelial growth factor secretion [3], differentiation into endothelial cells or pericytes [4] and vascular mimicry [5]. Clinically relevant, GSCs are resistant to both temozolomide and radiation exposure [6,7], and cell lineage studies have demonstrated that this population is responsible for tumor regrowth following treatment [8]. Therefore, for long lasting therapeutic responses, successful targeting and elimination of GSCs is necessary [9]. Although extensive research has identified several important cell signaling pathways and receptors responsible for GSC maintenance and many potential GSC targeted therapies have been developed, few have been effectively translated into clinical care [10,11]. Like conventional cytotoxic chemotherapies, these drugs need to cross the BBB at therapeutic levels with minimal off-target effects.

Nanoparticle-mediated drug delivery has the potential to cross the BBB and increase drug concentrations within the tumor and therefore increase treatment efficacy. To date most anti-cancer nanomedicines evaluated in clinical trials [12,13] rely on passive targeting due to the enhanced permeability and retention effect in which the nanomedicines accumulate in the tumors due to leaky vasculature and an impaired lymphatic drainage system. However, retention of nanomedicines is significantly impeded due to increased interstitial fluid pressure in brain tumors, which results in rapid removal of drugs from tumor extracellular space [14]. Nanomedicines need to be functionalized with BBB targeting and tumor-targeting ligands to improve delivery efficiency, tumor retention, and antitumor efficacy. Transferrin receptors (TFRs) are highly expressed at the BBB and facilitate the delivery of iron across the BBB and into the brain parenchyma by receptor mediated transcytosis. Similarly, GBM and cancer cells in general have increased levels of TFRs and transferrin conjugated nanoparticles increase tumor cell uptake by receptor mediated endocytosis [15]. Furthermore, since iron is essential for GSC maintenance and tumorigenesis, iron uptake and TFR cycling are increased in GSCs compared to the non-GSC cell population [16]. Therefore, using transferrin as part of the drug delivery system will target both GSCs and non-GSCs.

Non-toxic carbon-dots (C-dots) have emerged as excellent choices for developing drug delivery systems (DDS) and bioimaging agents due to their high biocompatibility, nano-size (<10 nm), photoluminescent properties, and versatility of surface functionality [17]. Derived from organic compounds, C-dots are chemically inert nanoparticles ideal for biomedical applications. Due to the large surface area to volume ratio and the presence of multiple functional groups, C-dots can be easily conjugated to inorganic and organic molecules. Furthermore, C-dots can cross the BBB [18,19] and preferentially accumulate in brain tumors [20]. We have previously demonstrated that C-dots chemotherapeutic conjugates can induce robust cell death in both adult and pediatric GBM cell lines [21,22,23]. In these studies, the chemotherapeutic agents (temozolomide, epirubicin, doxorubicin, and gemcitabine) are covalently attached to carboxyl groups present on the C-dot surface using the classical EDC/NHS coupling reaction. However, the formation of this amide bond requires the presence of an −NH_2_ group that is not readily available on most anti-cancer drugs. To overcome this obstacle, we synthesized a series of 4′-amino chalcones with the idea that the amino group would serve as a “handle” for conjugation to C-dots.

Chalcones are aromatic ketones belonging to the family of polyphenolic compounds. The anti-cancer properties of various natural polyphenolic compounds are well known [6,24,25], and have provided a diverse source of new medicinal leads [26]. Specifically, chalcones contain two phenyl groups and a 3-carbon aliphatic chain and form the foundation of many biologically active compounds. Both natural and synthetic chalcones have been shown to mediate anti-cancer effects in multiple types of cancer, including GBM. Chalcones have been shown to induce GBM cell death in vitro [27,28] and attenuate GBM tumor growth in vivo [29,30]. Furthermore, chalcones have been shown to inhibit specific cancer stem cell pathways responsible for cancer stem cell maintenance, proliferation, and viability [31,32]. For example, both Cardamonin, a 2,4-dihydroxy-6-methoxychalcone (also known as dihydroxymethoxychalcone), and Isoliquiritigenin, a 2′,4′,4-trihydroxychalcone, decreased GSC stem cell self-renewal and proliferation [33,34]. Taken together, these data support the development of chalcones to target GSCs. 

Therefore, the objective of this work was to synthesize a series of 4′-amino chalcones, test their cytotoxicity against a panel of GCS lines, and to investigate the potential of using C-dots as a nanocarrier to increase the anti-cancer effect. To study the role of the 4′-amino group in binding interactions (electrostatic vs. H-bonding), we synthesized N-acyl derivatives and 6-hydroxy derivatives of some of the more potent 4′-amino chalcones and tested the cytotoxicity in our cell lines. Lastly, we generated transferrin tethered C-dot-chalcone conjugates to investigate the enhancement of chalcone therapeutic efficacy by docking them into the tumor cell via the receptor mediated endocytosis. 

## 2. Materials and Methods

### 2.1. Chemistry

All reagents were obtained from commercial sources and were used without further purification. ^1^H- and ^13^C-NMR spectra were recorded at 500 and 125 MHz on Bruker. The spectra were referenced to the residual protonated solvents. Abbreviations such as s, d, t, m, br, and dd used in the description denote singlet, doublet, triplet, multiplet, broad, and double doublet, respectively. The chemical shifts and coupling constants were reported in δ parts per million (ppm) and hertz (Hz), respectively. High resolution mass spectra were obtained on Bruker micrOTO-Q II mass spectrometer. All intermediate and final products were monitored by thin layer chromatography (TLC) on 250 μm silica plates. Where applicable, the compounds were recrystallized from the proper solvent or purified by flash column chromatography on silica gel (200–300 mesh) with ethyl acetate/hexanes (1:1) as eluant. 

The syntheses of 4′-aminochalcones **3a**–**3x**, 4′-hydroxychalcones **4a**–**4f** and 3′-aminochalcone **5a** are outlined in Figure 1, Figure 2 and Figure 3, respectively. The chalcones were prepared by the Claisen–Schmidt condensation between the corresponding acetophenone derivative (1 equivalent) and the appropriate aryl aldehyde (1.1 equivalents) using either method a or b. The reaction was monitored by TLC. Upon completion, the reaction mixture was diluted with water and the resulting solid formed was collected by vacuum filtration. The chalcone was either purified by flash column chromatography or recrystallization. The purity of the synthesized compounds was analyzed by TLC with several solvent systems of different polarities. All the compounds were characterized by NMR (Nuclear Magnetic Resonance) and EI-HRMS (Electrospray Ionization-High Resolution Mass Spectrometry) analysis. Additional information on the synthesis (Appendix A) as well as the NMR and mass spectrometry data for the synthesized compounds (Appendix A) are provided in the Appendix A. The following chalcones have been previously synthesized and will not be included in the experimental section: chalcones **3a [35]**, **3b [35]**, **3c [36]**, **3e [35]**, **3f [35]**, **3i [37]**, **3k [38]**, **3l [39]**, **3n [35]**, **3o [40]**, **3p [38]**, **3q [41]**, **3r [41]**, **3s [42]**, **3t [43]**, **3u [41]**, **3v [41]**, **3w [41]**, and **3x [41]**, from the 4′-amino series; chalcones **4a [44]**, **4b [45]**, **4c [46]**, **4d [47]**, **4e** and **4f [48]**, from the 4′-hydroxy series; and chalcone **5a [49]**. MOM-protected derivative of vanillin was synthesized according to literature procedures [50]. 

### 2.2. Cell Culture

The generation and characterization of our GSC lines (Glio3, Glio9, Glio38) have been previously described [6,51,52]. In brief, following patient consent, the resected tumors were mechanically and enzymatically dissociated, and red blood cells were removed using Red Cell Lysis buffer (SigmaAldrich, St. Louis, MO, USA). Cells were filtered and plated in a 3:1 ratio of Dulbecco’s Modified Eagle’s medium (DMEM): F12 (Gibco, Carlsbad, CA, USA) media supplemented with 1% penicillin and streptomcycin (penn/strep), 20 ng/mL each of human epidermal growth factor and human fibroblast growth factor, and 2% Gem21 NeuroPlex Serum-Free Supplement (Gemini Bioscience, Sacramento, CA, USA); a formulation consistent for the generation of GBM stem cell lines. Both Glio3 and Glio38 were derived from tumors prior to drug or radiation therapy and grow as neurospheres, whereas Glio9 was derived from a recurrent tumor, following chemo and radiotherapy, and grows adherent. The GBM cell line U87 MG (U87), purchased from ATCC (Manassas, VA, USA) and the pediatric GBM cell line SJ-GBM2 was obtained from Children’s Oncology Group. SJ-GBM2 cell line was derived from a child with glioblastoma post therapy and is used as part of a National Cancer Institute pediatric pre-clinical testing program [53]. Both lines were maintained in RPMI media supplemented with 10% FBS and 1% penn/strep. Mesenchymal stem cells (MSCs) originally derived from human bone marrow aspirates were obtained from Thermo Fisher Scientific and maintained in MEM alpha supplemented with 20% FBS and 1% penn/strep. Prior to drug treatment, MSCs were moved to media with 10% FBS. Upon receipt U87, SJ-GBM2 and MSCs were immediately expanded with several aliquots frozen down for future use. These cell lines are not listed on the most recent NCBI database for misidentification and contamination of human cell lines. All cell lines were routinely tested for mycoplasma using LookOut mycoplasma PCR detection kit (Sigma-Aldrich, St. Louis, MO, USA) according to the manufacturer’s instructions and were maintained at 37 °C in a humidified 5% CO_2_ incubator.

### 2.3. In Vitro Drug Testing

The chalcones were dissolved in dimethyl sulfoxide (DMSO) at a concentration of 10 mM, vortexed and subsequently diluted to obtain 1 mM and 0.1 mM stock concentrations. C-dot-trans-chalcones were resuspended in supplement-free media (3:1 ratio DMEM/F12 for GSC experiments or RPMI for non-stem cell line experiments). Cell viability in response to drug treatment was determined using CellTiter 96 AQueous One Solution Cell Proliferation Assay, a colorimetric MTS assay kit (Promega, Madison, WI, USA). The GSCs were seeded into 96-well plates using modified neurosphere media containing 5% FBS at a density of 10,000 cells per well. MSCs and U87 cells were plated at 5000 cells/well while SJ-GBM2 cells were plated at 10,000 cells/well. The next day cells were treated with increasing concentrations of chalcones or C-dot-chalcone conjugate. Next, 72 h later, media was aspirated and 100 μL of a 1:5 solution of MTS to cell culture media was added to each well and incubated for 1–4 h. Optical density was measured at 490 nm using BioTek Synergy HT plate reader. Viability of drug treated cells is expressed as the percent viable cells relative to non-treated cells (100% viability). Experiments were done in triplicate.

### 2.4. Sphere Forming Assay

The effect of chalcones’ clonogenic growth potential was determined using sphere-forming assays. Glio3 and Glio38 neurosphere cell lines [6,7] were dissociated using Accutase (Gemini Bioscience, Sacramento, CA, USA), filtered using 40-micron filter and single cells were seeded at approximately 50 cells per well in a 96-well plate. Cells were treated with 0.1, 0.25 or 0.5 μM of chalcone or 0.01, 0.05 or 0.1 μM of C-dot chalcone conjugate on day 0 and spheres were manually counted under microscopy on day 14. All experiments were done in triplicate. Representative images were obtained using EVOS XL Core light microscope (Thermo Fisher Scientific, Waltham, MA, USA).

### 2.5. Western Blot Analysis

Our Western blot methods have previously been described [6,7,22,51,52]. Briefly, GSCs were harvested in RIPA buffer and proteins (20 µg) was separated using SDS–polyacrylamide gel electrophoresis and electroblotted to nitrocellulose membranes (Bio-Rad, Hercules, CA, USA). Membranes were probed with anti-TFRC (Cell Signaling Technology, Danvers, MA, USA) and anti-α-tubulin (Abcam, Waltham, MA, USA) and proteins visualized using enhanced chemiluminescence. 

### 2.6. Synthesis of C-Dots-Trans-Chalcone Derivatives

C-dots were synthesized from carbon-ash using an acid-oxidative top-down method commonly used in our group as previously described [54,55]. Briefly, C-dots (8 mg) were dissolved in 3 mL of 25 mmol/L phosphate buffer solution (PBS) (pH 7.4). Subsequently, 17.78 mg of 1-(3-dimethyl aminopropyl)-3-ethyl carbodiimide hydrochloride (EDC) (pre-dissolved in 0.5–1.0 mL of PBS) was added and the C-dots solution was stirred at room temperature for 20 min. *N*-hydroxysuccinimide (NHS) (10.68 mg/mL in PBS) was added, and after 20 min, 1 mL of transferrin (3 mg/mL in PBS) solution was added; 45 min later, 4.80 or 4.38 mg in 1 mL of 3,5-dimethoxy chalcone (**3f**) or 3,4,5-trimethoxy chalcone (**3e**) in DMSO was added to the C-dots reaction mixtures, respectively, and stirred overnight at room temperature. The mixtures were dialyzed using a 3500 molecular weight-cutoff dialysis membrane for 4 days, replacing the deionized water every 4–10 h. The reaction mixture was further purified by size-exclusion chromatography. The resultant eluent was collected into different aliquots and analyzed by ultraviolet–visible spectroscopy (UV-Vis) and fluorescence spectroscopies to identify the C-dots-trans-chalcone derivatives. The conjugates were frozen at −80 °C for 24 h and then lyophilized for 3 days to acquire the powdered sample.

### 2.7. Characterization of C-Dots-Trans-Chalcone Derivatives

The conjugates (20 μg/mL) were characterized by UV-Vis spectroscopy in a 1 cm quartz cell using Shimadzu UV-2600 spectrometer. The fluorescent emission spectra of the C-dots-trans-chalcone conjugates and the free chalcone were obtained by Horiba Jobin Yvon Fluorolog-3 with slit width 5 nm for both excitation and emission. The molecular weights of each C-dot-trans-chalcone derivative was analyzed with a Matrix-Assisted Laser Desorption/Ionization-Time of Flight (MALDI-TOF) mass spectrometer using a Bruker auto flex speed spectrometer.

### 2.8. Statistical Analysis

Significance was determined using Student’s *t*-tests for all pairwise comparisons of the different treatments that were tested. The results are presented as the mean  ±  standard error mean (SEM). Significance was set at *p*  <  0.05. Significance was determined by one-way analysis of variance (ANOVA) when comparing results of multiple chalcones together.

## 3. Results

### 3.1. Synthesis of Chalcone Analogues

The synthesis of two series of chalcone derivatives **3a**–**3x**, **4a**–**4f** and **5a** was performed according to Figure 1, Figure 2 and Figure 3, respectively. Initially, chalcones of the first series (**3a**–**3x**) were synthesized by Claisen–Schmidt reaction with yields of 27–97% after chromatography purification. The series of compounds has chalcones with amino at position 4′ (ring A) and benzene ring B substituted by electron-withdrawing (EWG) and electron-donating groups (EDG). Additionally, aryl analogues with benzene ring B replaced for furan and pyridine rings were synthesized (Figure 1). Chalcones with OH groups were prepared by using thionyl chloride as a catalyst in methanol under reflux. Compounds **3d** and **3g** were synthesized by reacting to the corresponding chalcones (**3c** and **3f**) with acetyl chloride and triethyl amine. Subsequently, another series of chalcone analogues (**4a**–**4f**) were synthesized. These analogues were prepared by a base-catalyzed Claisen–Schmidt condensation between 4′-hydroxyacetophenone and the appropriate aryl aldehyde with yields of 82% to quantitative after purification. The 4′-*O*-acetyl analogue (**4d**) was prepared by condensation of chalcone **4c** with acetyl chloride in the presence of triethylamine. Analogue **4e** was prepared by reacting chalcone **4c** with methyl bromoacetate under basic conditions. Hydrolysis of **4e** under basic conditions (LiOH/THF/MeOH/water) followed by acidification afforded the acid **4f**. More detailed description of the synthesis along with the spectral data for each compound can be found in the experimental section of the Appendix A.

### 3.2. 4′-Aminoacetophenone Series

The relative IC_50_s for the 4′-aminoacetophenone series is shown in Table 1. Of the 24 drugs examined in this group, only 10 (**3d**, **3f**, **3g**, **3o**, **3p**, **3q**, **3u**, **3v**, **3x**) had IC_50_ values below 10 µM in all GSC lines. Although chalcones **3c**, **3e** and **3w** were not as active, treatment of the GSCs did result in IC_50_ below 15 µM across all three cell lines. In general, these chalcones were relatively non-toxic to non-cancerous mesenchymal stem cells (MSCs) with a few exceptions. Notably, chalcones **3g** and **3p** demonstrated similar toxicity to both GSCs and MSCs. 

#### 3.2.1. Halogens

Some of the more effective chalcones were those containing halogens: chlorine, bromine or fluorine (**3u**, **3v**, **3w**, **3x**). Due to the potential for multiple interactions with proteins, halogens have found applications in medicinal chemistry [56]. Halogens have been widely used during the optimization phase of lead compounds, for example to enhance drug affinity and specificity. However, halogen binding can be exploited for lead discovery [57]. When averaged together, the IC_50_ for the 3 GSC was 8.6 µM, 7.2 µM, 9.1 µM, and 7.4 µM for chalcones **3u**, **3v**, **3w**, and **3x**, respectively. Interestingly distinct differences were observed between the different GSC lines. This is illustrated in Figure 1, which shows the percent viability of the different GSC lines in response to treatment with 10 µM of each halogen containing chalcone. It is clearly evident that Glio3 is less sensitive to the halogen substituted chalcones compared to Glio9 or Glio38. Halogens have been employed in the development of drugs to restore the function of the tumor suppressor p53 [57,58]. One such drug is nutlin-3 which contains chlorines and disrupts p53 binding with its inhibitor murine double minute 2 (MDM2). P53 pathway is often mutated in cancer, including GBM, and MDM2 is amplified in approximately 14% of patients [59]. Given our results, it is tempting to speculate that perhaps the p53 pathway is not altered in Glio3 or at least MDM2 is not amplified. Further studies are needed to confirm this hypothesis. The nutlin-3 derivatives Idasanutlin and SAR405838 contain both chlorines and fluorines and are currently in clinical trials for multiple cancers, however preclinical studies suggest poor ability to cross the BBB [60]. 

#### 3.2.2. Heteroaryl Chalcones

We investigated both furan (chalcone **3r** and **3s**) and pyridine (chalcone **3p** and **3q**) chalcone derivatives. Of these, only the pyridine derivatives were effective. While the position of the nitrogen seemed to play a minor role in the cytotoxicity toward GSCs, the average IC_50_ across all three cell lines of 6.1 ± 0.4 µM for the nitrogen at the 2 position (**3p**) and 7.2 ± 0.5 µM for the nitrogen at the 3 position (**3q**), the toxicity to normal non-cancer MSCs was highly different. Chalcone **3p** gave an estimated MSC IC_50_ of 8 µM compared to an estimated MSC IC_50_ of well above 10 µM for chalcone **3q**. These data indicate that **3q** rather than **3p** would be a better lead compound for further development. 

#### 3.2.3. Nitro Chalcone

The nitro group-containing compounds are used for a wide variety of clinical uses including anti-parasitic (e.g., Tinidazole), antitubercular (e.g., Delamanid), and anti-cancer (e.g., Pimonidazole) [61]. However, many nitro containing drugs are associated with severe toxicities, thereby limiting their therapeutic value. Our nitro derivative, chalcone **3o**, was one of the most effective chalcones tested, with an average IC_50_ of 6.04 ± 0.3 µM. Surprisingly, it was not very toxic with an estimated MSC IC_50_ of over 41 µM. In addition, unlike the other chalcones tested, chalcone **3o** significantly reduced neurosphere formation in both Glio3 and Glio38 at 250 nM concentration (Figure 2). This suggests that chalcone **3o** may be inhibiting cancer stem cell self-renewal pathways.

#### 3.2.4. Novel Chalcones 

We also synthesized five novel chalcones; three 4′-amino derivatives; 4-morpholino (**3m**), the 3-methoxy-4-methoxymethoxy (**3h**), 4′-amino-3 ethoxy-4-hydroxy (**3j**) as well as 2 *N*-acetylamino derivatives (**3d** and **3g**). Anti-cancer drugs containing morpholine interacting moieties include inhibitors of the PI3K/Akt/mTOR signaling pathway, perhaps one of the most commonly altered signaling pathways in cancer [62]. This is particularly true of GBM as mutations lead to hyperactivity of this pathway in up to 90% of GBM patients [63]. However, neither the morpholino or the methoxymethoxy derivatives were active against GSCs at the concentrations tested. These data suggest that large groups cannot be accommodated at the 4 position. Likewise, the 4′-amino-4-dimethyl amino (**3n**) was also inactive at the concentrations tested. The 4′-3-ethoxy-4-hydroxy chalcone (**3j**) was not active either at the concentrations tested. Based on our data, the presence of a hydroxy group at position 4 also appears to be deleterious for activity. Chalcones **3i**, **3j**, **3k** and **3h** all contain a hydroxy group at position 4 and were not active at concentrations tested. This is best illustrated by looking at chalcones **4e**, **4f** and **4k**. The 3,4,5-trimethoxy chalcone (**4e**) was active as was the 3,5-dimethoxy chalcone (**4f**) suggesting that the methoxy or any group at position 4 is not necessary for activity. However, substituting the methoxy group with a hydroxy group at position 4 (**4k**) eliminates activity. Both novel *N*-acetylamino chalcones, the 2,5-dimethoxy (**3d**) and the 3,5-dimethoxy (**3g**) were active and will be discussed in greater detail in the next section. However, it is important to note that while **3d** was only moderately toxic to MSCs with an IC_50_ of 16.8 µM, **3g** induced significant MSC death with an IC_50_ of 7.4 µM. 

#### 3.2.5. Methoxy Groups

In our previous work, we determined GSC cytotoxicity of curcumin-inspired bis-chalcones and found that methoxy groups were important for activity, with the greatest activity observed in the trimethoxy analog [51]. Therefore, we also examined both dimethoxy and trimethoxy chalcones. Similar to our previous results, the 3,4,5-trimethoxy chalcone **3e**, was effective in reducing GSC viability with an average IC_50_ of just below 10 µM. For the dimethoxy chalcones, the position of the methoxy group appeared critical for activity. This is illustrated in Figure 3A, in which cell viability in response to 10 µM of each dimethoxy chalcone is shown. The 4′-amino-dimethoxy chalcones inducing the highest toxicity were the 2,5 and 3,5-dimethoxy (**3c** and **3f**, respectively), whereas the 2,3 and 3,4 (**3a** and **3b**, respectively) were less effective, suggesting that the methoxy at position 5 is important for activity. Furthermore, the novel *N*-acetylamino 2,5 and 3,5-dimethoxy chalcone derivatives (**3d** and **3g**, respectively) were even more effective, indicating that the 4′ amino group is not involved in electrostatic interactions. In addition, this may suggest that a longer group at the 4′ position is increasing the interaction or strength of the drug with the binding pocket of the therapeutic target. As shown in Figure 3B, the average IC_50_ for the amino derivatives (**3c**, **3f**) is significantly higher than that of the acetylamino (**3d**, **3g**). To explore the effects of a better hydrogen-bond donor, we synthesized a series of 4′-hydroxy chalcones.

### 3.3. 4′-Hydroxyacetophenone Series and 3′-Amino Chalcone Derivatives

The IC_50_ for the 3′-amino trimethoxy chalcone and the 4′-hydroxyacetophenone series is shown in Table 2. Only 3 (**4b**, **4c** and **4d)** out of the 6 4′-hydroxyacetophenone chalcones tested had IC_50_ values below 10 µM. Furthermore, these were not as toxic to the non-cancer stem cell line. Conversely, the 3′-amino trimethoxy chalcone induced robust cell death in all 3 GSC lines as well as the MSC line. 

The 4′-hydroxy-2,3-dimethoxy analogue (**4a**) was more effective than the 4′-amino-2,3-dimethoxy (**3a**). Similarly, the 4′-hydroxy 3,5-dimethoxy analogue (**4b**) was more effective than the amino-3,5-dimethoxy (**3f**), which is illustrated in Figure 3A. In regard to the trimethoxy, the 4′-hydroxy (**4c**), 4′-acetoxy (**4d**) and 3′-amino (**5a**) were more effective than the 4′-amino trimethoxy (**3e**), which is illustrated in Figure 4B (viability) and Figure 4C (average IC_50_). Taken together, these data imply that the 4′-amino group is not required to interact via electrostatic attractions for activity and these modifications of the 4′-amino group or moving the amino group actually increases the efficacy of the chalcone. It is important to note that while moving the amino group to the 3′ position increased the toxicity to GSCs, it also dramatically increased the cytotoxicity to the non-cancer stem cells, MSCs. The MSC estimated IC_50_ for the 4′-amino-trimethoxy chalcone was close to 50 µM compared to an IC_50_ of 5.5 µM for the 3′-amino-trimethoxy chalcone. Substitution of the amino group with larger groups methoxycarboxymethyl and carboxylicmethyl (**4e** and **4f**) decreased the cytotoxicity towards GSCs possibly by altering either the binding strength, electrostatic repulsion (carboxylate group), or the target of the trimethoxy derivative. 

### 3.4. Chalcones Potential to Cross the Blood–Brain Barrier (BBB)

One of the greatest obstacles to effective GBM treatment has been the lack of effective drugs able to cross the BBB at therapeutic levels. In drug discovery, it is well-known that the physiochemical properties of the drug are important determinants of drug-likeness, or the likelihood of being orally active in humans. Lipinski’s rule of five (RO5) is a rule of thumb used in the development of drug-likeness [64]. The physiochemical properties include molecular weight (MW) < 500 kDa, lipophilicity or log*P* < 5, the number of hydrogen bond donors, ≤5, and hydrogen bond acceptors, ≤10. An additional criterion for oral activity was added by Veber et al.; the number of rotational bonds ≤ 10 [65]. However, unlike the gastrointestinal epithelium, the BBB is comprised of endothelial cells with tight junctions and astrocytic end feet significantly restricting the passage of solutes. Therefore, we examined the physiochemical properties associated with drugs that readily cross the BBB for chalcones with an IC_50_ below 10 µM in all GSC lines (Table 3). We included the polar surface area (PSA), as this relates too membrane permeability and is a good determinant for BBB permeability. The optimal values listed in Table 3 are based on an extensive evaluation of BBB penetrant drugs [66]. Overall the physiochemical properties of the active chalcones are indicative of drugs known to cross the BBB, with low MWs, limited number of H-bond donors and acceptors, reduced number of rotatable bonds and a low Log*P*. 

### 3.5. C-Dot-Trans-Chalcone Characterization

C-dots-trans-chalcone nano-motifs were employed as nano-carriers to increase the specificity and efficacy of the drug delivery system. C-dots were first conjugated with transferrin to facilitate cell membrane penetration via endocytosis. We chose the 3,4,5 trimethoxy (**3e**) and, the 3,5 dimethoxy (**3f**) chalcones to conjugate with C-dots as these promoted GSC death yet were relatively non-toxic to MSCs. The C-dots were conjugated to transferrin (trans) and to chalcones via covalent amide linkage (Figure 5). The structures of the C-dots-trans-chalcone conjugates were characterized by the UV-Vis and fluorescence spectroscopic techniques.

As shown in Figure 6, the transferrin conjugation on C-dots is indicated by the sharp peak overlap of the free transferrin and conjugates at 280 nm. The successful conjugation of dimethoxy on C-dots was suggested by the new band appearance at the 432 nm in the conjugate spectrum, which was red-shifted by 69 nm than the dimethoxy chalcone alone (Figure 6A). The presence of trimethoxy chalcone in the conjugate was noted by the peak overlap at 363 nm (Figure 6B). The fluorescence emission spectra confirmed the following findings (Figure 7). The conjugation of the 4′-amino-3,5-dimethoxy chalcone on C-dots was indicated by the wavelength-independent fluorescence emission peaks at 510 nm (Figure 7A), which was 10 nm red-shifted than the 4′-amino 3,5-dimethoxy chalcone (Figure 7B). Similarly, the conjugation of the 4′-amino-3,4,5-trimethoxy chalcone is indicated by the wavelength-independent fluorescence emission peaks at 511 nm (Figure 7C), which was 55 nm red-shifted compared to the 3,4,5-trimethoxy chalcone (Figure 7D). Thus, by considering the UV-Vis and fluorescence spectral data, the conjugation of both the chalcones on C-dots-trans systems can be verified.

### 3.6. Effect of C-Dot-Trans-Chalcone Conjugate on GSC Viability and GSC Self-Renewal

To determine if the C-dot-trans-chalcone conjugate could induce GSC death, cells were treated with increasing concentration ranging from 0.01 to 0.5 µM and resulting viability determined 72 h later. Prior to in vitro studies, C-dot-trans-chalcone conjugates were resuspended in cell culture media and a 0.1 mM stock was prepared based on the molecular weight of each, which is 85,000 g/mol. As shown in Figure 8, the dimethoxy (**3f**) conjugate was more effective than the trimethoxy (**3e**) conjugate for all cell lines. We previously demonstrated transferrin receptor expression in the GBM non-stem cell lines [19,20]. Similarly, our GSC lines demonstrate robust expression of the transferrin receptor (TFRC) as shown in Figure 8D.

However, both C-dot conjugates were significantly more effective than the free chalcone indicating that targeted C-dot mediated chalcone delivery increases its toxicity (Figure 9A). While the average GSC IC_50_ for the free trimethoxy (**3e**) was 9.7 ± 0.8 µM, the average GSC IC_50_ for C-dot-trans-**3e** conjugate was 0.42 ± 0.03 µM, representing about a 25-fold increase in GSC mediated cell death. The dimethoxy conjugate was even more effective, with an average GSC IC_50_ for the free dimethoxy (**3f**) of 8.2 ± 0.8 µM and the average GSC IC_50_ of 0.09 ± 0.02 µM for C-dot-trans-**3f** conjugate, indicating an approximate 100-fold increase in efficacy over drug alone (Figure 7A). Subsequently, we examined the effect of C-dot-trans-**3f** in two non-stem GBM cell lines, U87 (adult) and SJGBM-2 (pediatric). As shown in Figure 9B, the IC_50_ values of 3,5 dimethoxy chalcone (**3f**) for U87 and SJGBM2 cells were 8.5 ± 0.04 µM and 7.3 + 0.2 µM, respectively, while the IC_50_ values of C-dot-trans-**3f** for U87 and SJGBM2 were 0.12 ± 0.005 µM and 0.11 ± 0.009 µM, respectively. The fact that these values are similar to those observed for the GSC lines, suggests that both the free chalcone and the C-dot-trans conjugate can target both the GBM stem cell and non-stem cell population. Overall, the C-dot-trans-**3f** conjugate was about 75-fold more effective than the free chalcone in the non-stem cell lines, U87 and SJ-GBM2. 

To determine the effect of C-dot-trans conjugate **3f** on GSC self-renewal; sphere-forming assays were performed. Glio3 and Glio38 cells were treated with c-dots alone, transferrin alone or 10, 50 and 100 nM of the C-dot-trans-**3f** conjugate and the number of neurospheres counted 10 days later. As shown in Figure 10, neither c-dots or transferrin alone significantly reduced neurosphere formation when compared to non-treated controls. However, the 10 nM concentration significantly reduced neurosphere formation in Glio3, while 50 nM concentration completely prevented neurosphere formation in both cell lines. 

## 4. Discussion

Despite extensive research aimed at understanding the molecular drivers of GBM as well as thousands of clinical trials worldwide, there has been little improvement in patient outcome in decades. In recent years, it has become evident that treatment resistant GSCs contribute to the poor outcome in GBM. GSCs drive tumor growth, contribute to tumor heterogeneity and are responsible for tumor recurrence. Targeting these cells has become a major area of research. 

Both natural and synthetic chalcones have been shown to decrease the malignant characteristics of cancer stem cells including GSCs. The aim of this study was to examine the potential of targeting GSCs using small molecule chalcones. Of the 31 chalcones synthesized and tested, 14 (10 in the 4′-aminoacetophenone, 3 in the 4′-hydroxyacetophenone series as well as the 5′-amino chalcone derivative) resulted in IC_50_ values of about 10 µM or below in all 3 GSC lines. Furthermore, several of these chalcones were relatively non-toxic to non-cancerous human MSCs. In general, the halogen containing chalcones (**3u**, **3v, 3w** and **3x**) demonstrated good activity against the GSCs with only moderate toxicity toward MSCs. Of the heteroaryl chalcones, the pyridines (**3p**, **3q**) rather than the furans (**3r**, **3s**) were more active, however the position of the nitrogen greatly affected the toxicity toward the MSCs. The most effective chalcone synthesized was the nitro chalcone (**3o**), which has an average GSC IC_50_ of about 6 µM. Furthermore, the nitro chalcone was the only chalcone to significantly reduce GSC neurosphere formation at sub-toxic concentrations, indicating the ability to target GSC stem cell self-renewal pathways. Lastly, the nitro chalcone was minimally toxic to normal cells, with an MSC IC_50_ of 41 µM. Methoxy containing chalcones were also effective against GSCs with the following caveats. The 3,4,5-trimethoxy (**3e**) was effective, however, substituting a hydroxy at position 4 abrogated activity (**3k**). For the dimethoxy chalcones, the position of the methoxy groups is important for activity, with the 2,5 and 3,5 (**3e**, **3f**) dimethoxy chalcones being more effective than the 2,3 or 3,4 (**3a**, **3b**), suggesting that the methoxy at the 5 position is important for activity. The N-acetylamino 2,5 and 3,5-dimethoxy chalcone derivatives (**3d**, **3g**) were more effective than their 4′-amino counterparts (**3e**, **3f**) indicating that the 4′-amino group is not involved in electrostatic interactions. Similarly, the 4′-hydoxy derivatives (**4b**, **4c**) were more effective than their 4′-amino counterparts (**3f**, **3e**). However, the hydroxy and N-acetylamino derivatives were also more toxic to MSCs. Taken together, these data suggest that the 4′-amino group is not required to interact via electrostatic attractions for activity. Using the amino group as a handle for conjugation we then conjugated two chalcones (**3e**, **3f**) to transferrin tethered C-dots via the amide linkage and demonstrated that this C-dot DDS was up to 100-fold more efficacious than the free chalcone in our GSC lines and in non-stem GBM cell lines (U87 and SJGBM2). 

The ease of 4′-amino chalcone synthesis and their straightforward conjugation to C-dots makes them excellent lead candidates for the development of GSC-targeted anti-GBM therapies. C-dots have emerged as an exceptional choice for a DDS especially for delivery across the BBB owing to their small size, easy functionalization, and high biocompatibility [23,67,68,69]. Furthermore, C-dots can increase specificity and efficacy via specific targeting ligands which increase cellular uptake of the chalcones by multiple mechanisms including, but not limited to, receptor mediated endocytosis. Targeted chalcone C-dot conjugates have the potential to induce robust tumor cell death and increase the survival for patients currently facing a dismal prognosis.

## 5. Conclusions

In summary, we designed, synthesized and evaluated three series of 31 chalcones against glioblastoma stem-like cells (GSC, Glio3, Glio9, and Glio38). Among the 31 chalcones, five were novel chalcones (**3d**, **3g**, **3h**, **3j**, and **3m**), two of which demonstrated activity against GSCs. Overall, we identified 14, which induced robust GSC death with IC_50_ values of about 10 µM or below in all three GSC lines. Furthermore, many of these chalcones were relatively non-toxic to non-cancerous human MSCs. The BBB severely limits the efficacy of systemic treatment as most anti-cancer drugs cannot cross the BBB at therapeutic levels. However, the physiochemical properties of these 14 compounds suggest the ability to cross the BBB, although additional studies are required. The results suggest that several of the chalcones may serve as lead compounds for further structural optimizations to find more potent and less toxic derivatives for treatment of GBM alone or as nanoparticle formulation via covalent attachment.

## Data Availability

All data generated is presented in the publication.

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
