# Peer review of "Chalcones as Anti-Glioblastoma Stem Cell Agent Alone or as Nanoparticle Formulation Using Carbon Dots as Nanocarrier"

_pharmaceutics, 2022, doi:10.3390/pharmaceutics14071465_

Round 1

Reviewer 1 Report

This manuscript by  Graham et al. reports scholarly carried out research and significant results. It deserves to be published with minor modifications, possibly considering the following points.

  1. It is unclear to me how the authors determine the relative ratio of ferritin and chalcone in the doubly functionalized C-dots. I also don’t understand how the concentration of functionalized C-dots is determined. This point is obviously related to the previous one. Perhaps I have missed something in the manuscript. In any case, this information needs to be properly highlighted in the experimental section.
  2.  Less important is that the authors when referring to the synthesis of C-dots (line 205) do not cite the paper reporting it but subsequent papers referring to this previous paper (Chem. Mater. 2015, 27, 5, 1764–1771). This way of proceeding is rather annoying. 

Author Response

Reviewer number 1.

Comments and Suggestions for Authors

This manuscript by  Graham et al. reports scholarly carried out research and significant results. It deserves to be published with minor modifications, possibly considering the following points.

We thank you for your suggestions and the opportunity to improve our manuscript.

  1. It is unclear to me how the authors determine the relative ratio of ferritin and chalcone in the doubly functionalized C-dots. I also don’t understand how the concentration of functionalized C-dots is determined. This point is obviously related to the previous one. Perhaps I have missed something in the manuscript. In any case, this information needs to be properly highlighted in the experimental section.

We neglected to include the information. The molecular weight (MW) of each C-dot-transferrin-chalcone conjugate was determined by MALDI-Mass spectroscopy. The MW of each conjugate was 85 kda and millimolar stock solutions were prepared based on this. I have added this to the text. See sections 2.7 and 3.6.

  1. Less important is that the authors when referring to the synthesis of C-dots (line 205) do not cite the paper reporting it but subsequent papers referring to this previous paper (Chem. Mater.2015, 27, 5, 1764–1771). This way of proceeding is rather annoying. 

       My apologies. We have added this reference and removed 2 of the other references. Section 2.6.

Reviewer 2 Report

The article entitled  Chalcones as anti-glioblastoma stem cell agent alone or as nanoparticle formulation using carbon dots as nanocarrier is a document of interesting subject matter.

1. It is expected to have an extensive literature review followed by an in-depth and critical analysis of the state of the art, and identify challenges for future research in the Introduction.

2. The authors should do the analysis the conclusion section must clearly establish a strong correlation with the proposed topic.
3. Your abstract should clearly state the essence of the problem you are addressing, what you did and what you found and recommend. That will help a prospective reader of the abstract to decide if they wish to read the entire article

4. The objective or objectives should be clearly elucidated in the last paragraph of the introduction.

6. Try to create a purposeful relationship between paragraphs in the Introduction.

Author Response

Reviewer number 2.

The article entitled  Chalcones as anti-glioblastoma stem cell agent alone or as nanoparticle formulation using carbon dots as nanocarrier is a document of interesting subject matter.

Thank you for your suggestions and the opportunity to improve our manuscript. We have greatly changed the writing of the paper to improve the content and flow of the paper. Please see all the track changes.

  1. It is expected to have an extensive literature review followed by an in-depth and critical analysis of the state of the art, and identify challenges for future research in the Introduction.

We have added additional background information and references to the introduction

  1. The authors should do the analysis the conclusion section must clearly establish a strong correlation with the proposed topic.

We have added to the conclusion

  1. Your abstract should clearly state the essence of the problem you are addressing, what you did and what you found and recommend. That will help a prospective reader of the abstract to decide if they wish to read the entire article

We have greatly improved the abstract to increase clarity.

  1. The objective or objectives should be clearly elucidated in the last paragraph of the introduction.

We now clearly state the objectives in the last paragraph of the introduction.

  1. Try to create a purposeful relationship between paragraphs in the Introduction.

We improved the transition from one paragraph to another in the introduction.

Round 2

Reviewer 2 Report

Authors addressed all comments carefully.